# Preliminary Findings from the Gulf War Women’s Cohort: Reproductive and Children’s Health Outcomes among Women Veterans

**DOI:** 10.3390/ijerph19148483

**Published:** 2022-07-11

**Authors:** Alexa Friedman, Patricia A. Janulewicz Lloyd, Jeffrey Carlson, Emily Quinn, Dylan Keating, Rosemary Toomey, Timothy Heeren, Steven S. Coughlin, Glenn Markenson, Maxine Krengel, Kimberly Sullivan

**Affiliations:** 1Department of Environmental Health, Boston University School of Public Health, Boston, MA 02215, USA; paj@bu.edu (P.A.J.L.); jmcarls5@bu.edu (J.C.); dmk13@bu.edu (D.K.); tty@bu.edu (K.S.); 2Biostatistics and Epidemiology Data Analytics Center, Boston University School of Public Health, Boston, MA 02118, USA; eq@bu.edu; 3Department of Psychological and Brain Sciences, Boston University, Boston, MA 02118, USA; toomey@bu.edu; 4Department of Biostatistics, Boston University School of Public Health, Boston, MA 02118, USA; tch@bu.edu; 5Department of Population Health Sciences, Medical College of Georgia, Augusta University, Augusta, GA 30912, USA; scoughlin@augusta.edu; 6Charlie Norwood VA Medical Center, Augusta, GA 30912, USA; 7Department of Obstetrics & Gynecology, Boston University School of Medicine, Boston, MA 02118, USA; glenn.markenson@bmc.org; 8Department of Neurology, Boston University School of Medicine, Boston, MA 02118, USA; mhk@bu.edu

**Keywords:** Gulf War, veterans, women, reproductive health, children’s health

## Abstract

Reproductive outcomes, such as preterm birth, miscarriage/stillbirth, and pre-eclampsia, are understudied in veterans, particularly among Gulf War veterans (GWVs). During deployment, women GWVs were exposed to toxicant and nontoxicant exposures that may be associated with adverse reproductive and developmental outcomes. The data come from a survey of 239 participants from northeastern and southern U.S. cohorts of women veterans. The questionnaire collected information about the service history, current and past general health, reproductive and family health, demographic information, and deployment exposures. Odds ratios were computed with 95% confidence intervals between exposures in theater and reproductive/children’s health outcomes. GWVs experienced adverse reproductive outcomes: 25% had difficulty conceiving, and 31% had a pregnancy that ended in a miscarriage or stillbirth. Pregnancy complications were common among GWVs: 23% had a high-risk pregnancy, and 16% were diagnosed with pre-eclampsia. About a third of GWVs reported their children (38%) had a developmental disorder. Use of pesticide cream during deployment was associated with higher odds of all reproductive and developmental outcomes. The results demonstrate that GWVs experienced reproductive and children’s health outcomes at potentially high rates, and exploratory analyses suggest pesticide exposure as associated with higher odds of adverse reproductive outcomes. Future longitudinal studies of women veterans should prioritize examining reproductive and children’s health outcomes.

## 1. Introduction

Thirty years post deployment, veterans of the 1991 Gulf War (GW) continue to experience several adverse health outcomes [1,2,3], especially when compared to veterans not deployed to the GW [3,4,5]. Specifically, GW veterans (GWVs) experience an increased prevalence of musculoskeletal, neurological, pulmonary, gastrointestinal, and dermatological symptoms [6,7], most notably as the result of toxicant exposures experienced during deployment. In total, these symptoms have been termed Gulf War Illness (GWI) [2,8], a chronic multisymptomatic illness.

Research has documented that both men and women veterans have an increased prevalence of health symptoms, and there is some evidence that suggests women, compared to men, suffer from worse health outcomes [9,10,11]. At the time of the GW, approximately 49,000 women served, which represented the largest proportion of women serving in a war zone in United States military history at the time [10,11]. Despite the increased number of women who served during the GW, few studies on veteran’s health have specifically focused on women’s health outcomes, including reproductive health outcomes related to their service.

Recent studies have focused on sex-specific health symptoms and medical conditions, and some studies have found a larger prevalence of GWI and GWI-related health symptoms in women veterans compared to male veterans [12,13,14,15]. A 2020 study of prevalence and patterns of health symptoms among women veterans by Sullivan et al. found deployment to the GW was associated with more adverse health symptoms compared to women who served during the same time but were not deployed to the GW [11]. In 2008, reproductive health was made a priority by the Congressionally Directed Research Advisory Committee on Gulf War Veterans’ Illnesses (RAC-GWVI), a committee which is charged to provide advice to the Secretary of Veterans Affairs on the nature and scope of research related to veteran health [2,12].

Outcomes such as adverse pregnancy, birth, and children’s health outcomes are generally understudied in veterans and particularly for those who served in the GW [2,12]. Studies have found evidence for an increased rate of all birth defects associated with deployment to the Persian Gulf region during the GW [16,17,18], and there are data to suggest a possible association of GW service with serious birth defects, such as congenital abnormalities [19]. For example, in a study of over 75,000 infants born between 1991 and 1993 to male veterans, the authors found that children born to GW veterans, were three times as likely to have Goldenhar Syndrome, compared to the children of non-GW veterans [20]. Similarly, in a study of over 2.3 million veterans, children born to GW male veterans were between two to six times more likely to have congenital heart defects [18]. Further, a previous study found that in couples of GW-deployed veterans (either men or women deployed), there was an increased risk of miscarriage and/or stillbirth compared to non-GW deployed veterans [21]. This suggests that deployment to the GW may be related to other reproductive outcomes; yet, there is limited research in this area. 

There are even fewer studies examining the relationship between GW military service and the health of offspring. To our knowledge, only one published study has examined the overall health of the children of GW veterans, and in an exploratory analysis, this study found that children of GW veterans experienced worse dentition, greater rates of obesity, and more behavioral problems compared to non-deployed veterans’ children [22]. Thus, further investigation of veterans of the GW and their reproductive health and their children’s health is critical to beginning to address this research gap. 

We report preliminary findings of two regional surveys from the Gulf War Women’s Cohort study research group, a group designed to specifically address questions on women’s health following deployment [12]. First, we report the prevalence of reproductive outcomes among women who served in the U.S. military during 1990–1991 and describe trends in the reproductive health outcomes of women veterans deployed to the GW (GWVs) compared to non-GW-deployed women who served during the same era (GW-era). Second, we examined the risk of reproductive outcomes following multiple exposures experienced by deployed GWVs. Our exploratory findings contribute to the limited research on reproductive outcomes among women veterans and their children’s health.

## 2. Materials and Methods

### 2.1. Study Population

The Gulf War Women’s Cohort study conducted two regional surveys of women veterans from northeastern and southeastern states. The same survey was given to both survey groups and was designed to specifically address questions on women’s health following deployment, including questions about general health, mental health outcomes, and reproductive outcomes [12]. The survey methods have been previously published [12,23]. Briefly, for the Southern survey, the survey was conducted through Augusta University, and the Northeastern survey was conducted through Boston University School of Public Health. All study participants served in the military during 1990–1991 (i.e., the GW-era). The study population consisted of 239 participants from northeastern and southeastern cohorts of women veterans (Figure 1).

The lead site for the Gulf War Women’s Cohort was Augusta University, and women were recruited at multiple sites if they had participated in previous GW studies. Informed consent was obtained from all subjects involved in the study. The Institutional Review Boards at the Boston University Medical Campus and Augusta University approved all study protocols.

### 2.2. Data Collection

Participants were first contacted by telephone. Once participants provided consent, the phone interviewer proceeded with the survey questions, which were recorded using REDCap web-based electronic data capture. For respondents who were unable to be reached by phone, data were collected using postal survey questionnaires developed in a TeleForm optical character recognition system. A detailed description of the methods is available in prior publications [12,13]. The questionnaires asked about demographics, service history, current and past general health, reproductive and family health, and questions about health status and health history to identify GWI status [12,23].

### 2.3. Deployment Status and Case Definitions

Women were considered GW-era veterans if they were deployed during 1990–1991 to an area not within the Persian Gulf (PG) region and were classified as GW veterans if they reported having been deployed during 1990–1991 to the PG. To be defined as a GWI case (Kansas Case Definition), veterans met multiple or moderate-to-severe chronic symptoms in at least three of six statistically defined symptom domains: fatigue/sleep problems, somatic pain, neurological cognitive, mood symptoms, gastrointestinal symptoms, respiratory symptoms, and skin abnormalities [4]. To be defined as a GWI case by the Center for Disease Control (CDC), a participant had 1 or more chronic symptoms (present for ≥6 months) from at least 2 of the following categories: fatigue; mood and cognition (symptoms of feeling depressed, difficulty remembering or concentrating, feeling moody, feeling anxious, trouble finding words, or difficulty sleeping); and musculoskeletal (symptoms of joint pain, joint stiffness, or muscle pain) [5]. Case status for GWI is only applicable among GWVs (i.e., those deployed to the Persian Gulf); thus, we did not evaluate case status for women who were deployed elsewhere (i.e., GW-era).

### 2.4. Reproductive and Children’s Health Outcomes

Reproductive health outcomes from the surveys described above included difficulty conceiving, having had a miscarriage/stillbirth, having been told you were a high-risk pregnancy, having hypertension during pregnancy, and being diagnosed with pre-eclampsia. Children’s health outcomes included having had a child born prematurely, having a child with a birth defect, and having a child with a disability. Women were included in the evaluation of pregnancy outcomes if they had reported having tried to become or having been pregnant. Women were included in the evaluation of children’s health outcomes if they reported having had a child (Figure 1).

### 2.5. Exposures in Theater

In the second part of our analyses, we examined the association between theater exposures from GW deployment and reproductive outcomes; therefore, only women who had been deployed to the PG were included (Figure 1). Women were excluded if the reported reproductive outcome occurred before deployment (i.e., 1991) to ensure temporality between deployment and outcome (N = 26). Several GW-deployment-related exposures were examined including both neurotoxicant and non-neurotoxicant exposures (Appendix A). Exposures during deployment included seeing troops who had been badly injured or killed or encountering a destroyed enemy vehicle, hearing a chemical alarm sounding, using a pesticide cream or liquid, taking pyridostigmine bromide (PB) pills (i.e., anti-nerve pills), and/or sleeping in a tent with a propane space heater. Respondents were asked about the frequency of exposures during deployment (no exposure, exposure for 1–6 days, exposure for 7–30 days, or greater than 30 days). Given our sample size, we consolidated the exposure groups: GWVs were considered unexposed if they reported no exposure or exposure less than 7 days and were considered exposed if they reported an exposure for longer than or equal to 7 days.

### 2.6. Statistical Analysis

Frequency distributions and cross-tabulations of the data were computed overall as well as stratified by area of deployment (i.e., to PG [GWV] or to another location during the same time [GW-era veteran]). A small number of women reported having a child with a birth defect (N = 4; 4.6%); therefore, we did not report on this outcome. In the subset of women who were deployed to the PG (GWVs), we examined the association between exposures in theater and reproductive outcomes in an exploratory analysis. The exposure prevalence is listed in Appendix A. We computed odds ratios with 95% confidence intervals, when possible (based on sample size).

All statistical analysis were using R version 3.6.1

## 3. Results

### 3.1. Demographics

A total of 101 women veterans reported having tried to become pregnant and/or having been pregnant (Figure 1). Of those women, 87 had a child, which represents the sample from which we examined children’s outcomes (Figure 1). A total of 77 GW-deployed women veterans reported having tried to become pregnant/or having been pregnant, and a total of 63 GW-deployed women veterans reported having a child (Figure 1).

The mean age of the participants in both subsets and by deployment status ranged from 55.7 to 57.6 years old at the time of survey (Table 1). The women veterans were majority white, though the distribution of women who were white was different between GW and GW-era veterans (Table 1). Among women veterans who reported having tried to become pregnant, 81.8% GWVs were white compared to 50.0% of GW-era veterans. In addition, in the pregnancy subset, the majority of GWVs reported being currently married or cohabitating (54.7%) compared to 40.0% of GW-era veterans. Most veterans, regardless of group, had either a four-year degree or higher (Table 1). Veterans of the GW had an equal distribution of household incomes whereas the GW-era veterans tended to have higher household incomes. Of those who were deployed to the PG (GWVs), nearly all the women veterans (>93%) were identified as having GWI by both the Kanas and CDC definition (Table 1).

### 3.2. Prevalence of Reproductive and Children’s Health Outcomes by Deployment

The prevalence of all reproductive outcomes is reported in Table 2. Among all veterans, 21 (20.8%) reported having difficulty conceiving, and the rate was higher among those deployed to the PG, of whom 24.7% reported difficulty conceiving compared to 8.3% in GW-era veterans. Among all veterans, 26 (25.7%) reported pregnancies that ended in miscarriage or stillbirth. This rate was higher among GWVs compared to GW-era veterans (31.2% vs. 8.3%), but there was a high portion of missing data on this question among GW-era veterans (70.8% vs. 5.2% missing for GWVs). One-fifth of women veterans reported they were told by a medical provider that they were a high-risk pregnancy (20.8%), and the rate of this was higher in those deployed to the GW (23.4% vs. 12.5%). A similar pattern was observed for women reporting pregnancy complications such as pregnancy hypertension and pre-eclampsia (Table 2).

Of the 87 women who reported having children, 63 were GWVs and 24 were GW-era veterans. Overall, 13.8% of veterans reported having a preterm birth, and this rate was similar among GWVs compared to GW-era veterans (14.3% vs. 12.5%, respectively). One-third of women veterans (36.8%) reported that they had a child with any type of disability (e.g., learning difficulties, hyperactive disorder, or frequent behavioral problems). The rate of disabilities was similar between GWVs and GW-era veterans, where 38.1% of GWVs and 33.3% of GW-era veterans reported their child had a disability.

### 3.3. Associations between Exposures in Theater and Reproductive and Children’s Health Outcomes

In an exploratory analysis, we calculated the odds of an adverse reproductive and children’s outcome following wartime exposures. We saw evidence that pesticide cream use was associated with higher odds of all reproductive outcomes, though the results were imprecise (Table 3). For example, using pesticide cream for at least 7 days during deployment was associated with 5 times the odds of having a child with any disability compared with using pesticide for less than 7 days or not using pesticide cream at all (95% CI: 1.23, 25.93). There were no consistent patterns between exposures with other reproductive and children’s health outcomes (Appendix A). However, seeing injured troops during deployment was associated with increased odds of having a pregnancy that ended in a miscarriage/stillbirth [OR = 3.42 (95% CI: 1.15, 10.81)] (Appendix A). Lastly, we observed an association between sleeping in a tent with a propane space heater for more than 7 days and pregnancy-related hypertension [OR = 4.41 (95% CI: 1.06, 20.05)] (Appendix A).

## 4. Discussion

Among women veterans deployed to either the PG or another area during 1990–1991, there was evidence of potentially high rates of adverse reproductive and children’s health outcomes. The rate of all pregnancy complications, including being told you were a high-risk pregnancy, pregnancy hypertension, or pre-eclampsia was higher among GWVs compared to GW-era veterans, as well higher than the rates in the general population. For example, the Centers for Disease Control and Prevention (CDC) reports that pre-eclampsia occurs 1 in 25 pregnancies (~4%), and the rate in other research studies examining the association between environmental exposures and pre-eclampsia has varied between 2 and 8% [24,25], whereas we report that 10% of women veterans reported having had pre-eclampsia. The rate of having a child with any disability was twice as high among all subgroups compared to the rate reported by the CDC, which is about ~15% (or 1 in every 6 children) [26]. Our exploratory analyses also showed that exposures in theater were associated with higher odds of adverse reproductive and children’s health outcomes. Taken together, this preliminary report suggests that women veterans, both GWVs and GW-era, may have been more susceptible to adverse reproductive and children’s health outcomes.

It has been well established that deployment to the GW is associated with several unique exposures that have had long-lasting effects [1,2,3,7,11]. Common exposures examined in previous studies of GW veterans include exposure to pyridostigmine bromide bills (e.g., anti-nerve pills), pesticides, sarin, and mustard gas, all of which may result in increased physiological oxidative stress [1]. Studies have also found GW veterans were at a higher risk of mental health diagnoses, including posttraumatic stress disorder, major depressive and other depressive disorders, as well as anxiety disorders [9]. We observed that nontoxicant exposures, such as having seen troops badly injured, were also associated with increased odds of adverse pregnancy outcomes. This suggests that exposures, even nontoxicant ones, during military service can lead to adverse reproductive outcomes. This may be due to, in part, the long-lasting effects of psychological stress on the body, resulting in increased physiological stress (e.g., oxidative stress), which has been shown to be related to changes in reproductive health. For example, oxidative stress can impact reproductive organs, such as the ovaries, ultimately affecting oocyte count and quality [27,28].

The health of deployed veterans’ offspring is generally understudied, although one study found worse dentition, greater obesity, and more behavioral problems among children from deployed parents [22]. However, the findings from Toomey et al. were also exploratory in nature, as the children included in their study were only examined when veterans who were part of another study brought their child with them unprompted, which led to the potential for selection bias [22]. Regardless, both the findings from our exploratory analysis as well as Toomey et al. (2021) affirm the need for research focused on the health of children of veterans. Military service, GW-related exposures, and/or GWI may also impact child development through parenting styles. Preexisting health conditions have been associated with changes in parenting behaviors such as being demanding and responsiveness. In turn, modification of the parent–child relationship has been shown to influence child adjustment [29]. Our findings suggest that there is a potential relationship between exposure during military service and developmental health outcomes in offspring, which warrant further attention in future studies.

The aim of this preliminary report was to explore the association between deployment and deployment exposures with reproductive and children’s health outcomes. This area of research was made a research priority by the Veteran Affairs Research Advisory Committee on Gulf War Veterans’ Illnesses report entitled Gulf War Illness and the Health of Gulf War Veterans [2] but remains largely understudied. For example, to date, there is no study that has specifically addressed whether reproductive, birth, and children’s health outcomes impact veterans with GWI differently than those without GWI. While we were specifically interested in veterans of the GW, the results here and in subsequent studies may have implications for all women veterans, past and present. By understanding how service may impact reproductive health, research can begin to elucidate possible mechanisms and, therefore, interventions. Similarly, future studies focused on the potential transgenerational effects of military service and exposures would be the first step in identifying vulnerable groups and thus resources for children of veterans.

While our sample size was limited, this preliminary report highlights the need for more research examining the relationship between deployment/exposures in the GW and reproductive outcomes. While self-reported survey data have their own limitations (e.g., may be subject to outcome misclassification), the use of self-reported survey data, as opposed to other studies that have mainly relied on medical records, can potentially capture more accurate outcome data specific to reproductive outcomes such as difficulty conceiving and/or miscarriage/stillbirth. For example, the survey collected data on difficulty in conceiving using a yes or no question. This outcome may not be always captured on medical records unless the parent sought medical treatment (e.g., in vitro fertilization [IVF]) or was deemed medically sterile. On the other hand, we were unable to standardize across participants for some outcomes, such as for difficulty conceiving or being told you were a high-risk pregnancy, as the questions were broad. Thus, we may be capturing a range of reasons or meanings for these outcomes and broad definitions, which may be one possible explanation for why we see high rates in our study. A further limitation was that we were unable to differentiate between miscarriage and stillbirth or the types of pre-eclampsia. This may particularly be important for miscarriage and stillbirth, which are two distinct reproductive outcomes with potentially different biological mechanisms.

All respondents have been part of other GW studies; thus, there is the potential that we would observe higher pathology, as this is a studied cohort. We were also not able to look at the effect of having GWI, since nearly all veterans in our study were identified as having GWI (~93–96%). The rate of GWI was higher in our survey than what has been reported in other larger studies of GWVs (the average prevalence is between 25 and 32%) [2], although some studies have suggested this rate is higher among women [10,11]. Thus, there is some concern for selection bias, where women with GWI were more likely to participate in our survey than women without GWI. However, the prevalence of GWI among all women from entire the survey population (N = 239) was 61.5%, which was lower than our analytical sample (N = 87–101), which does not necessarily remediate the concern for selection bias; however, it does suggest that it is possible that adverse reproductive outcomes are more common among cases, than controls, although we did not have the ability to examine that in this study.

Despite the limitations, there are only a few studies that have investigated reproductive outcomes [11,18,30,31] and only one that examined the health of GW veterans’ children [22]. We observed that deployment to the Gulf War and subsequent exposures during deployment may be related to adverse reproductive and children’s health outcomes. Future studies should continue to explore this relationship and consider including additional reproductive outcomes, including cancers of reproductive organs. Even though our sample size was small (n = 63–101 women veterans, depending on subset), the study results provide valuable insights and suggest that the association between deployment, exposures in theater, and reproductive and children’s health outcomes should be further studied.

## 5. Conclusions

The results suggest that research exploring the relationship between military deployment, exposures in theater, and reproductive and children’s health outcomes should remain a priority in military research.

## Figures and Tables

**Figure 1 ijerph-19-08483-f001:**
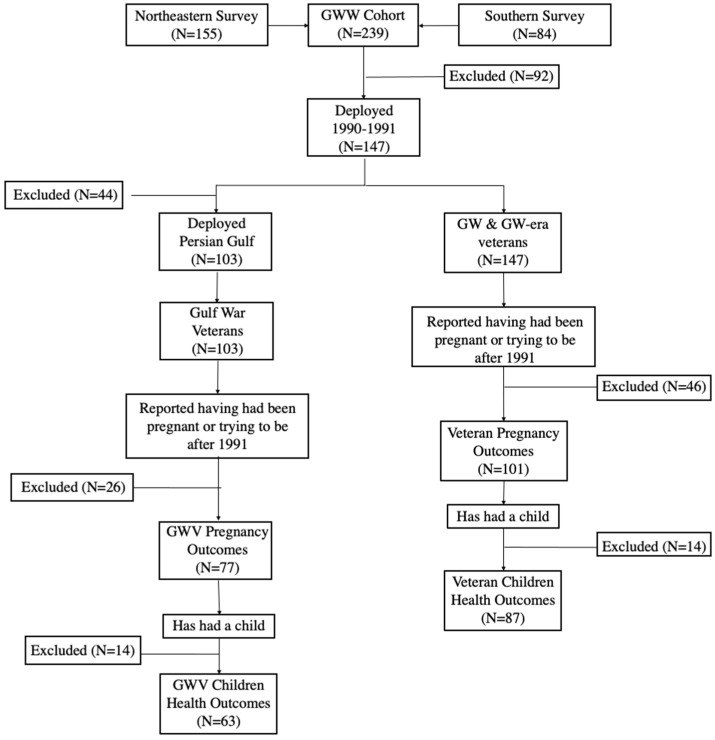
Schematic of the study population.

**Table 1 ijerph-19-08483-t001:** Selected sociodemographic characteristics of women veterans.

	Pregnancy Outcomes (N = 101)		Children’s Health Outcomes (N = 87)	
	GWV (N = 77)	GW-Era (N = 24)	GWV (N = 63)	GW-Era (N = 24)
**Age** (years), mean [sd]	55.7 [7.0]	57.6 [5.9]	55.8 [7.2]	57.5 [6.4]
**Race**				
White or Caucasian	63 (81.8%)	12 (50.0%)	49 (77.8%)	13 (54.2)
Black or African American	10 (13.0%)	11 (45.8)	9 (14.3%)	10 (41.7%)
Other	2 (2.6%)	1 (4.2%)	3 (4.8%)	1 (4.2%)
**Highest Education Obtained**				
High school/GED	3 (3.9%)	1 (4.2%)	2 (3.2%)	1 (4.2%)
<4-year degree	31 (40.3%)	9 (37.5%)	26 (41.3%)	8 (33.4%)
≥4-year degree	43 (55.8%)	43 (55.8%)	34 (54.0%)	14 (58.4%)
**Relationship Status** (current)				
Married/ cohabitating	47 (54.7%)	10 (40.0%)	36 (57.1%)	10 (41.7%)
Divorced/separated	22 (24.4%)	14 (56.0%)	19 (30.2%)	13 (54.2%)
Widowed	5 (5.8%)	1 (4.0%)	5 (7.9%)	1 (4.2%)
Single/never married	11 (12.8%)	0 (0.0%)	2 (3.2%)	0
**Household Income Level**				
≤50,000	27 (35.5%)	4 (16.6%)	19 (30.1%)	4 (16.6%)
50,000–75,000	20 (26.0%)	5 (20.8%)	16 (25.4%)	5 (20.8%)
≥75,000	25(32.9%)	12 (50.0%)	22 (35.0%)	7 (29,2%)
Prefer Not to Answer	4 (5.2%)	2 (8.3%)	4 (6.3%)	2 (8.3%)
**GWI Case Status**				
Kansas Criteria	72 (93.5%)	-	59 (93.7%)	-
CDC Criteria	74 (96.1%)	-	60 (95.2%)	-

**Table 2 ijerph-19-08483-t002:** Prevalence of reproductive outcomes, by deployment status.

	All Respondents	GW-Deployed	GW-Era
**Reproductive Outcomes**	N = 101	N = 87	N = 24
Difficulty conceiving	21 (20.8%)	19 (24.7%)	2 (8.3%)
Pregnancies ended in miscarriage/stillbirth	26 (25.7%)	24 (31.2%)	2 (8.3%)
High risk pregnancy ^a^	21 (20.8%)	18 (23.4%)	3 (12.5%)
Pregnancy hypertension ^a^	14 (13.9%)	12 (15.6%)	2 (8.3%)
Pre-eclampsia ^a^	11 (10.9%)	9 (11.7%)	2 (8.3%)
**Children’s Health**	N = 87	N = 63	N = 24
Child born preterm	12 (13.8%)	9 (14.3%)	3 (12.5%)
Child with any type of disability ^b^	32 (36.8%)	24 (38.1%)	8 (33.3%)

^a^ Told by physician or medical provider ^b^ Includes child hyperactivity disorder, frequent behavioral problems and/or other learning disabilities reported by respondent.

**Table 3 ijerph-19-08483-t003:** Association between using pesticide cream/liquid on the skin during deployment and the odds of adverse reproductive and children’s health outcomes among women deployed to the Persian Gulf.

	Exposure Status ^c^	Odds Ratio (95% CI)
Difficulty conceiving	Exposed	2.33 (0.69, 9.39)
	Unexposed	1.00
Pregnancies ended in miscarriage/stillbirth	Exposed	1.88 (0.55, 7.60)
	Unexposed	1.00
High risk pregnancy ^a^	Exposed	2.43 (0.70, 10.03)
	Unexposed	1.00
Pregnancy hypertension ^a^	Exposed	**8.10 (1.35, 156.16)**
	Unexposed	1.00
Pre-eclampsia ^a^	Exposed	4.96 (0.78, 97.19)
	Unexposed	1.00
Child born pre-term	Exposed	3.82 (0.58, 75.81)
	Unexposed	1.00
Child with any type of disability ^b^	Exposed	**5.00 (1.23, 25.93)**
	Unexposed	1.00

^a^ Told by physician or medical provider ^b^ Includes child hyperactivity disorder, frequent behavioral problems and/or other learning disabilities reported by respondent. ^c^ Exposed represents ≥7 days of exposure; unexposed represents no exposure or <7 days of exposure.

## Data Availability

The data presented in this study are available on request from the corresponding author. The data are not publicly available due to privacy concerns.

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
