# Peer review of "Preliminary Findings from the Gulf War Women’s Cohort: Reproductive and Children’s Health Outcomes among Women Veterans"

_ijerph, 2022, doi:10.3390/ijerph19148483_

Round 1

Reviewer 1 Report

The report titled ’Preliminary findings from the Gulf War Women’s Cohort: reproductive and children’s health outcomes among women veterans’ written by Alexa Friedman et al. is a well-written manuscript, it draws people’s care to this special population, women veterans. Especially in current time, where war is really happening, it is meaningful to point out its hazards to women soldiers and their offspring in the future. The overall study design, data analysis and results interpretation are plausible and rational, but few questions need to be considered and answered.

1.     In table 1 about the demographics, in all these characteristics of the GW, is any of them statistically different from that of the GW-era? The same question is raised up for table 2. This is important to clarify to conclude that the observed phenomenon is only a trend or with significant difference.

2.     Why does the sample size of GW-era group so small comparing to that of the GW group?  Additionally, as pointed out in line 188 and 189, there is a high portion of missing data in GW-era group, which further weakened the power of the results and conclusions. How will this be solved in this manuscript? Or if this is preliminary analysis, how will it be solved in the formal analysis?

3.     In the comparison in table 3, do they all belong to the GW group? Please clarify it.

4.     I would like to add following points in the discussion section:

The Gulf War happened 30 years ago, the final aim of retrospectively analyzing its hazards to women veterans should be to provide something for current and future women in military to avoid or reduce the risk of undergoing such bad effects. So, what are the common factors that possibly exist in today’s military life that women facing, or in future wars? How could these be avoided or reduced? If they have already been exposed to those factors, is there any solutions or later care could help to rescue or reduce this impact?

Author Response

Thank you for your feedback. Please see the attachment for response.

Reviewer 2 Report

The paper is interesting and in accord with the literature is quite complete.

Contribution of all authors is significant and could be interesting for scientist.

Analysis and data interpretation are adequacy, writing style could be improving.

The only suggestion is you could add a drawing this it would make reading the manuscript more appealing.

Moreover, I suggest improving how a tailed and multidisciplinary approach is needed by citing:

Buonomo B, Massarotti C, Dellino M, Anserini P, Ferrari A, Campanella M, Magnotti M, De Stefano C, Peccatori FA, Lambertini M. Reproductive issues in carriers of germline pathogenic variants in the BRCA1/2 genes: an expert meeting. BMC Med. 2021 Sep 10;19(1):205. doi: 10.1186/s12916-021-02081-7

Author Response

Thank you, please see the attachment.

Reviewer 3 Report

Grammar and Syntax

Abstract: No edits

Introduction: No edits. Very clearly written.

Materials and Methods:

-Line 110-111 :- maybe should put proprietary information (company, city) for REDCap and Teleform in brackets. Why both? Was one at Boston and one at Augusta? 

 Line 159: there seems to be a word missing: “in an exploratory…”

Conceptual

Materials and methods

General question: was there a human operator doing the survey or was it all automated? If there was a human operator was the operator blinded to exposure group (GWV versus GW-era)

-Line 104: a bit unclear: was the study population identified bases on a database of women who had participated in prior studies?  Maybe that could create a bias for higher pathology because this is a studied cohort. (Am I understanding correctly). Maybe a bit more clarity on how the study (and control) subjects were collected

Section 2.4- 128-136: Maybe make it clearer at this point if the survey looked at all pregnancies, and the outcomes of all  pregnancies in each of the women or looked at whether they ever had one of the complications such as PE, stillbirth etc in all pregnancies cumulative. This becomes clearer with further reading (seems cumulative) but it may be easier to read if clarified upfront. It may have been good to know the total number of pregnancies per group and the average number of pregnancies per participant for each of the groups.

Line 149- Maybe clarify what PB pills are because some readers won’t be familiar 

Results:

Table 1: I note there are 24 controls for both groups; is there a design reason for why all the controls who wanted to have children had children?

I note the percentage of people with GWI is very high.. How was GWI ascertained? Were the specific criteria items asked on the survey or were the respondents asked if they met criteria?

Would it not have been good to see how many GW-era respondents met GWI criteria as well (esp since the prevalence is so high)

Table 3 a lot of comparisons were made: I am assuming “crude” means no post-hoc correction for multiple comparisons. Is this right?

Line 204” “evidence that pesticide cream use was associated 204 with higher odds of all reproductive outcomes”…can you say this given that many of the OR confidence intervals cross 1.

Discussion:

225- I think it may not be “fair” to compare the incidence of medically diagnosed preeclampsia with self-reported preeclampsia but I know I’m quibbling. Your point is good: the rates of adversity are too high (colloquially and informally but still its important! I agree!!)

258- Just a general observation – in my experience surveys often overestimate incidences and medical chart or record type studies often underestimate incidences. Nothing seems to get it totally right!

262: just to quibble “eclampsia is not a type of preeclampsia (its not “pre” anymore because of a seizure or neurological symptoms). Toxemia is just an old fashioned synonym for preeclampsia. You can get away with this by just deleting the brackets because there are many types or at least a broad spectrum of PE

264- I would like if you speculated why almost all had GWI in your study

Overall

Your work is important and has a lot of merit and needs to be disseminated.

Reviewer 4 Report

A brief summary

This study by Friedman et al. adds to the limited research interrogating the reproductive health of female veterans of the Gulf War and the developmental health of their children. In addition to comparing health outcomes of veterans deployed to the Persian Gulf or deployed elsewhere, it also offers comparisons of demographic factors. Furthermore, the study delineates pathophysiology of toxicant and non-toxicant exposures with regards to outcomes, including robust definitions of Gulf War Illness pertaining to the studied cohort. In the face of limited research on the topic, the literature review is complete and relevant. While some aspects of the study are not clearly explained or defined, these limitations do not detract from its overall utility and relevance to the field.

General concept comments with specific potential reviews

Overall, the piece is clear, well-structured, and easy to follow. The topic is relevant as women continue to be involved in combat and exposed to chemical and psychological stressors. A strong case is made for the lack of research on the topic currently, especially for the developmental health of the children of Gulf War veterans.

The methodology is of an appropriate design to answer the research question, and easily reproducible. While the authors do make an adequate argument for the use of a self-report questionnaire, some arguments remain unclear—for example, on line 259, they state that self-report questionnaires “may capture more accurate outcome data specific to reproductive outcomes like miscarriage/stillbirth” without elaborating how.

While the study does report on an admirable range of reproductive health outcomes, some of the studied outcomes are unclear in their definition. For one, the authors do not comment on the difficulty of standardizing the research participants being labelled as high-risk pregnancies by their healthcare providers. Otherwise, the outcome of “difficulty conceiving” is not adequately defined—line 184 seems to allude to this being a self-reported measure as compared to being confirmed by the HCP, but this is not made clearer neither earlier in the piece nor later.

With regards to toxicant and non-toxicant exposures, there is again an admirable range of exposures covered, as well as common-sense operationalized definitions of these exposures, such as hearing a chemical alarm sound. However, the authors used a cut-off of 7 days of exposures to these stressors; this 7 day cut-off was not explained either by the authors or by a citation. Additionally, it is unclear whether this cut-off refers to every listed stressor—this begs the question of whether the participants required 7 days of hearing chemical alarm sirens to qualify as having been exposed to chemical weapons the same as needing to spend 7 days or more in a propane-heated domicile (another of the measured stressors). With regards to some of the lack of clarity of toxicant exposures, this confusion is mitigated by the use of both the Kansas and CDC case criteria for Gulf War Illness (which is attributed to toxicant exposure).

The provided tables, both in the study itself and the supplemental material are clear and easy to understand, apart from the aforementioned definitions of “difficulty conceiving” or “high-risk pregnancy”.

Review: The authors have done well to use sources that are fairly recent (10-15 years old), relevant, and complete on a topic that is minimally covered and specific to the time and place of the Gulf War. However, due to the smaller breadth of the literature pertaining to the topic, it may have been worthwhile to explore the literature in more depth. For example, the study by Toomey et al. (22) is cited as the only example of literature interrogating the health of Gulf War veterans’ children, specifically. However, there is nothing written about the limitations of this study, even though it is the only source of information about the developmental aspect of the study done by Friedman et al. While it may be too much to critique each study cited, it may be helpful to readers to know that the findings of Toomey et al about Gulf War veteran children were exploratory in nature, and were not originally meant to be measured at all. They were only measured when veterans brought their children to the studies without prompting, introducing a potential selection bias.
